# Positive Echocardiographic Association between Carotid Artery and Coronary Artery Diameter and Z-Score in a Mouse Model of Kawasaki Disease

**DOI:** 10.3390/diagnostics14020145

**Published:** 2024-01-08

**Authors:** Wen-Ling Shih, Tsung-Ming Yeh, Kuang-Den Chen, Steve Leu, Ho-Chang Kuo

**Affiliations:** 1Department of Biological Science and Technology, National Pingtung University of Science and Technology, Neipu 912301, Taiwan; wlshih@mail.npust.edu.tw (W.-L.S.); ytm@mail.npust.edu.tw (T.-M.Y.); 2General Research Service Center, National Pingtung University of Science and Technology, Neipu 912301, Taiwan; 3Kawasaki Disease Center and Department of Pediatrics, Kaohsiung Chang Gung Memorial Hospital, Kaohsiung 83301, Taiwan; dennis8857@gmail.com; 4Institute for Translational Research in Biomedicine, Kaohsiung Chang Gung Memorial Hospital, Kaohsiung 83301, Taiwan; 5Department of Biotechnology, College of Life Science, Kaohsiung Medical University, Kaohsiung 83301, Taiwan; 6College of Medicine, Chang Gung University, Taoyuan 33302, Taiwan

**Keywords:** Kawasaki disease, Z score, coronary artery, carotid artery, animal model, *Lactobacillus casei*

## Abstract

Kawasaki disease (KD) occurs in young children, has an unknown etiology, and can cause such life-threatening complications as coronary artery aneurysm. A mouse model using *Lactobacillus casei* cell wall extract (LCWE) with intraperitoneal injection was established for KD years ago. Histological examination of coronary artery lesions indicated features similar to those of vascular lesions of patients with KD. Since animals must be sacrificed during histological examination, the longitudinal survey of coronary artery lesions (CALs) is difficult. The aim of this study was to survey the vasculitis status of the coronary artery and the carotid artery in a KD mouse model. Method: LCWE was intraperitoneally injected into 5-week-old male C57BL/6 mice to induce CALs. We studied the longitudinal status of the carotid and coronary arteries and analyzed the Z-score of coronary artery diameter. Results: Carotid artery wall thickness (day 7) and diameter (day 14) significantly increased in the LCWE group with a dose-dependent effect (*p* < 0.05). Aortic diameter and wall thickness demonstrated significant increases on day 28 and day 7, respectively (*p* < 0.05). Carotid artery outer diameter and wall thickness were positively associated with coronary artery diameter on day 28 (*p* < 0.01). Coronary artery diameter significantly increased in the LCWE group after day 7 (*p* < 0.05). The percentage of Z > 3.0 indicated was more than 80% in the high-dose LCWE group and 0% in the control group. Conclusions: This report is the first to use coronary artery Z-score in a mouse model of KD by echocardiography and to find a positive association between carotid artery and coronary artery diameter.

## 1. Introduction

Kawasaki disease (KD) was first reported more than 50 years ago by Dr. Tomisaku Kawasaki in 1967 in Japan. Nevertheless, the etiology and pathogenesis remain unknown and continue to be investigated. KD is among the most common systemic vasculitis to occur in children aged 2–5 years old, following only Henoch–Schoenlein purpura. KD is diagnosed according to the five major clinical symptoms and signs suggested by the American Heart Association (AHA). While early treatment with intravenous immunoglobulin (IVIG) and aspirin has significantly decreased the rate of coronary artery lesions (CALs), KD-related coronary arteritis and aneurysm formation are still the most common causes of acquired heart disease in children in developed countries [1].

Mouse models have been widely used to study the etiology of vasculitis in KD including studies that involved the administration of a *Candida albicans* water-soluble fraction (CAWS), a *Lactobacillus casei* cell wall (LCWE) extract, and a synthetic Nod1 ligand. The vasculitis observed in these murine models is similar to the CALs of patients with KD, especially with regard to inflammatory cell infiltration [2]. Animal models provide an opportunity to study the progression of KD in a controlled environment, with mouse models playing a crucial role in the pathogenesis and mechanisms of CALs in KD. By inducing similar symptoms and observing disease development, researchers can gain insights into the biological processes and immune responses involved. Mouse models allow for genetic manipulations, such as knockout or overexpression of specific genes, in order to investigate their roles in KD. For example, Kocatürk et al. reported that platelets exacerbate cardiovascular inflammation in a murine model of KD vasculitis [3]. Lin et al. also reported that not only may Toll-like receptor (TLR) 2 augmentation on CD14+ monocytes be an inflammatory marker for both human KD patients and LCWE-induced CAL mice, but that this model may also be feasible for studying therapeutic strategies for coronary arteritis in human KD by modulating TLR2-mediated immune activation on CD14+ monocytes. That same study also reported that the macrophage dectin-1/spleen tyrosine kinase (Syk)-mediated pathway is involved in the LCWE-induced CAL mouse model and in the production of interleukin-6 (IL-6) and MCP-1 [4].

This research has helped to identify the genetic factors involved in disease susceptibility and progression. Mouse models can be used to assess the efficacy and safety of potential therapeutic interventions for KD. By administering drugs or treatments to mice with similar disease characteristics, researchers can evaluate their impact on disease outcomes. Furthermore, mouse models enable researchers to conduct in-depth mechanistic studies, including the examination of immune responses, inflammatory pathways, and vascular changes associated with KD. Using experimental mouse models of KD vasculitis has considerably improved our understanding of the pathology of the disease and helped characterize the cellular and molecular immune mechanisms that contribute to cardiovascular complications, thus aiding the development of innovative therapeutic approaches [5]. Findings from mouse models can be transferred to human studies to provide clinical benefit. The aim of this study was to establish an echocardiographic image survey of the carotid artery and coronary artery in a KD mouse model to provide longitudinal coronary artery evaluation.

## 2. Materials and Methods

### 2.1. Animals

Wild-type (WT) male C57BL/6 mice purchased from the National Laboratory Animal Center in Taiwan were used in this study. All animal experiments were performed in accordance with legislation on the protection of animals and were approved by the animal care committee at Chang Gung Memorial Hospital with certificate No. 2022081801. To induce CAL- in the mouse model, 4–5-week-old WT mice were intraperitoneally injected with 1 mL of PBS containing 1 mg of LCWE or with 1 mL of PBS alone as a negative control group. Echocardiography was performed on the mice at 7, 14, 21, and 28 days post-injection (n = 9 at each time point). Mice were sacrificed on day 28, at which point plasma samples were collected for cytokine detection.

### 2.2. LCWE Preparation

LCWE was prepared as described in a previous study (Lehman et al., 1985) [6]. Briefly, cells of *L. casei* (ATCC 11578), purchased from the Bioresource Collection and Research Center, Taiwan, were first cultured in Lactobacillus MRS broth (Difco, Detroit, MI, USA) at 37 degrees Celsius. After being harvested, the cells were treated overnight with 4% SDS (Sigma–Aldrich, St. Louis, MO, USA) and then sequentially incubated with 250 μg/mL RNase, DNaseI, and trypsin (Sigma–Aldrich). The final pellet was then sonicated (5 g packed wet weight in 15 mL PBS) for 2 h at a pulse setting of 9 s pulse/5 s pause at a 20 kHz frequency (Vibra CellTM, Sonics & Materials Inc., Newtown, CT, USA). Following 1 h centrifugation at 20,000× *g*, the supernatant concentration was determined based on its rhamnose content by using a phenol–sulfuric acid colorimetric assay, as in our previous report [4].

### 2.3. High-Resolution Echocardiography and Z-Score

LCWE was prepared and injected into C57BL/6 mice intraperitoneally to induce vasculitis of KD. All mice underwent two-dimensional echocardiography of the coronary artery and high-resolution B-mode ultrasonography of the common carotid arteries. The mice were positioned in a supine orientation, as illustrated in the flow chart, and an ultrasonic coupling agent was applied to their chest wall. The left sternum was chosen for long- and short-axis evaluations using ultrasonography, employing a high-resolution small animal ultrasound system, as previously reported by Zhang et al. [7] The left main coronary artery (LCA) was measured midway between the ostium and the bifurcation of the circumflex artery and the left anterior descending coronary artery in the parasternal short-axis view. The same senior technician performed all high-resolution echocardiography detections of the mouse hearts. Animal echocardiography was performed by measuring the coronary artery dimensions, cardiac function, and carotid artery on days 0, 7, 14, 21, and 28 (high-resolution small animal ultrasound (Vevo 3100, FUJIFILM VisualSonics Inc. Toronto, ON, Canada). *Z*-scores were calculated with means and standard deviations from the data collection at day 0 by using regression with the square root of body surface area. *Z*-scores beyond the normal limits (cutoff of *Z* > 3.0) were considered abnormal.

### 2.4. Statistical Analysis

The experimental results are presented as mean ± standard error of the mean (SEM). To compare the two groups, we employed a two-tailed Student’s *t*-test. The vessel diameter was adjusted by body weight by dividing the diameter by body weight and then multiplying by 100. Pearson correlation was utilized to determine the statistical significance among multiple comparisons. The threshold for statistical significance was set at *p* ≤ 0.05.

## 3. Results

### 3.1. Sonography Findings of the Coronary and Carotid Arteries

The flowchart of this study is presented in Figure 1, illustrating the key components such as the animals involved, the timeline of the study, and the echography procedures. As shown in Figure 2, echocardiography of the LCWE group’s (intraperitoneal injection 2 mg/kg for 7 days) coronary artery diameter on day 28 after study was significantly greater than that of the control group (no LCWE treatment). The normal range of coronary artery diameter was found to be 0.217 ± 0.028 mm (N = 18), while the outer coronary artery diameter was 0.472 ± 0.056 mm (N = 37). Sonography of carotid artery wall thickness in the LCWE group on day 28 after the study showed a significant increase compared to the control group (Figure 3). Carotid artery wall thickness significantly increased in the LCWE group with the narrowing of the internal diameter when compared with the control group.

### 3.2. Carotid Artery and Aortic Artery in KD Mouse Model

Carotid artery wall thickness (outer diameter minus inner diameter, day 7) and diameter (day 14) significantly increased in the LCWE group and demonstrated a positive dose-dependent effect (*p* < 0.05) with LCWE dosage. Carotid artery wall thickness increased significantly in the LCWE group when compared with the control group in a dose-dependent manner (*p* < 0.05, Figure 4). Aortic arch outer diameter demonstrated a significant increase on day 28, as did wall thickness on both day 7 and day 28 (*p* < 0.05, Figure 5). The changes observed in the carotid artery and aortic artery in LCWE mice suggest that the vasculitis in the KD mouse model demonstrates systemic involvement.

### 3.3. Coronary Artery Diameter Significantly Increased in LCWE Mice

Coronary artery diameter showed a significant increase in LCWE mice from day 7 with a dose-dependent effect (*p* < 0.05, Figure 6). A higher dose of LCWE resulted in a higher percentage of coronary artery dilation. The ejection fraction decreased in the LCWE group on day 21. Carotid artery outer diameter (*p* = 0.004) and wall thickness (*p* = 0.005) were positively associated with coronary artery diameter on day 28 (Figure 7). The outer diameter and wall thickness of the carotid artery could serve as a useful tool for predicting coronary artery dilatation.

### 3.4. Z-Score in KD Mouse Model

*Z*-score was set up using all mice (*N* = 38) on day 0 before any treatment and calculated with the mean to obtain the standard deviation (SD) by using regression with the square root of body surface area. This study marks the first time the *Z*-score has been analyzed in a KD mouse model from the literature review. *Z*-score can more precisely detect coronary artery lesions in a mouse model. The percentage of *Z* > 3.0 (3.0 SD greater than the mean value) was more than 80% with a high dose and 50% with a medium dose in the LCWE group and was 0% in the control group from day 21 to day 28 (Figure 8). When a *Z*-score > 3.0 was applied in this mouse model of KD, more than 80% of CALs were detected after day 21 without sacrificing any animals.

## 4. Discussion

KD animal models provide various advantages for clinical research, particularly the development of new therapeutic agents and investigation into disease mechanisms. According to a previous report, the *Z*-score is a standardized score that represents how much a value is above or below the population mean in a normally distributed population [8]. In the present study, *Z* > 3.0 accounted for more than 80% in the LCWE group but 0% in the control group from day 21, thus providing a more precise way to identify and monitor CAL in KD mice rather than sacrificing the mice. Furthermore, the echocardiographic Z-score may also provide longitudinal CAL follow-up after intervention or treatment, which the traditional sacrifice method cannot do.

KD is an inflammatory illness of unknown etiology that is often complicated by coronary artery inflammation. The mouse model of CAL has been widely used as an animal model of KD for decades [6]. Systematic study of the pathologically similar CAL induced in mice by a single intraperitoneal injection of LCWE fragments may provide useful clues to not only the pathogenesis of KD but also appropriate treatment [9]. Sacrificing mice in order to collect blood and tissue samples for investigation is currently the most common method used in KD animal studies. Pathological studies have revealed that focal inflammatory cells infiltrate around the coronary artery of the KD model. Taking advantage of high-resolution small animal ultrasound, an echocardiographic study that includes coronary dimension, carotid artery, aortic arch, and cardiac function measurements could be successfully performed in C57BL/6 mice with a body weight of only 20 g (i.e., 1/2000 of a human weighing 40 kg).

IVIG treatment within the first 10 days after disease onset of KD is highly effective for the acute phase of this illness and considerably reduces the prevalence of CALs. Therefore, early recognition and precise treatment of KD are crucial. Measurement of intima-media thickness (IMT) of common carotid arteries is a widely used and validated noninvasive imaging technique for assessing early structural changes in the arterial wall [10]. Wu et al. [10] reported that the mean carotid IMT in KD patients (0.550 ± 0.081 mm; range, 0.44–0.69 mm) was significantly higher than that in the febrile control children (0.483 mm ± 0.046 mm; range, 0.43–0.56 mm; *p* = 0.01). The IMT during the acute stage of KD is increased, suggesting that IMT could be a valuable tool in the early stages of KD. The IMT of the carotid artery and the diameter of the coronary artery were found to be higher in KD patients, both with and without coronary artery lesions (CALs), compared to normal controls. Additionally, the IMT of the KD patients with CALs was observed to be higher than that of the KD patients without CAL [11]. Inflammation of vessel walls with morphological changes is believed to be the initial presentation of such rheumatic diseases as ankylosing spondylitis, Behçet’s disease, Takayasu’s arteritis, systemic lupus erythematosus, and rheumatoid arthritis. Previous reports revealed that in long-term follow-up, carotid artery IMT is greater in KD patients and even greater in KD patients with CALs [12,13]. Patients with KD have an increased aortic IMT and reduced carotid distensibility, indicating greater cardiovascular risk, especially in those with coronary artery abnormalities. In our study, we used validated surrogates for cardiovascular disease risk [14] and found that carotid artery outer diameter and wall thickness are positively associated with coronary artery diameter, results that are compatible with echocardiographic findings in children with KD. A carotid artery survey could provide early recognition of KD and detection of CAL formation for aggressive treatment.

A limitation of this study is the lack of correlation between ultrasonographic results and pathological findings, which could provide deeper insights into the reasons for the increased wall thickness of the carotid and coronary arteries. To enhance understanding, future evaluations should extend to human subjects to determine if there is an association between the carotid and coronary arteries. Additionally, due to the small number of cases in this study, further investigations involving a larger cohort of either animals or KD patients are needed to validate and generalize the findings.

## 5. Conclusions

This report is the first to utilize coronary artery Z-score assessment in a mouse model of KD through echocardiography. It provides a non-invasive method for monitoring coronary artery lesions with a higher rate of detection and offers the advantage of longitudinal follow-up. The positive association observed between the carotid artery and coronary artery diameter offers a simpler method for surveying coronary artery lesions (CALs) through the carotid artery. Conducting a survey of the carotid artery could prove to be a useful means of identifying KD and detecting the formation of coronary artery lesions.

## Figures and Tables

**Figure 1 diagnostics-14-00145-f001:**
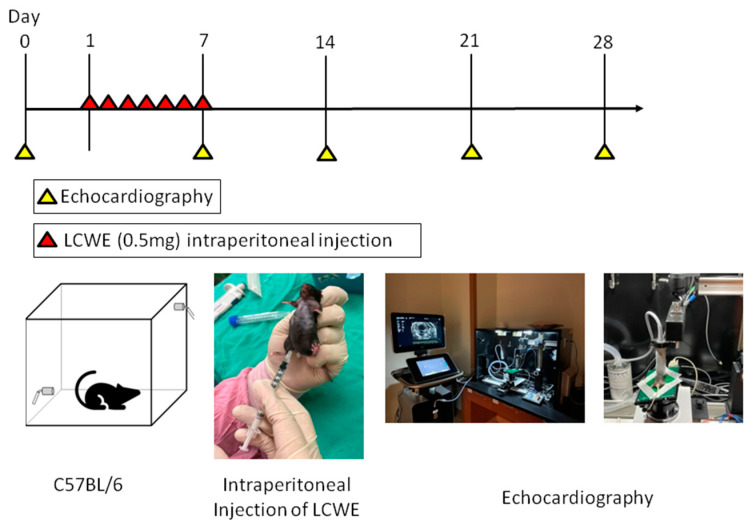
Study flow chart.

**Figure 2 diagnostics-14-00145-f002:**
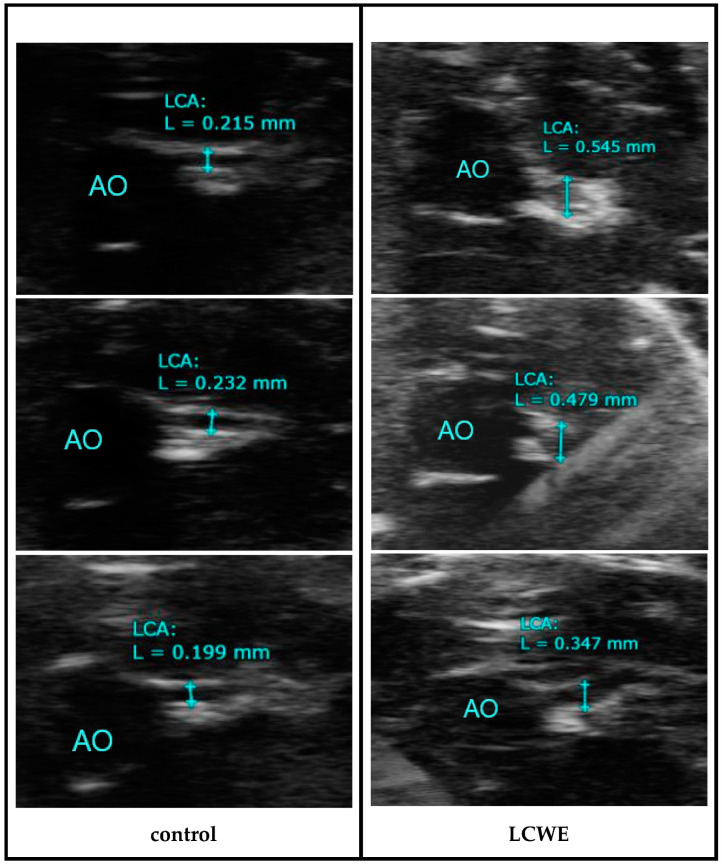
Coronary artery diameter in the control group (no treatment) and LCWE group (intraperitoneal injection 2 mg/kg for 7 days) by echocardiography on day 28 after study (AO: aorta, LCA: left main coronary artery, L: diameter).

**Figure 3 diagnostics-14-00145-f003:**
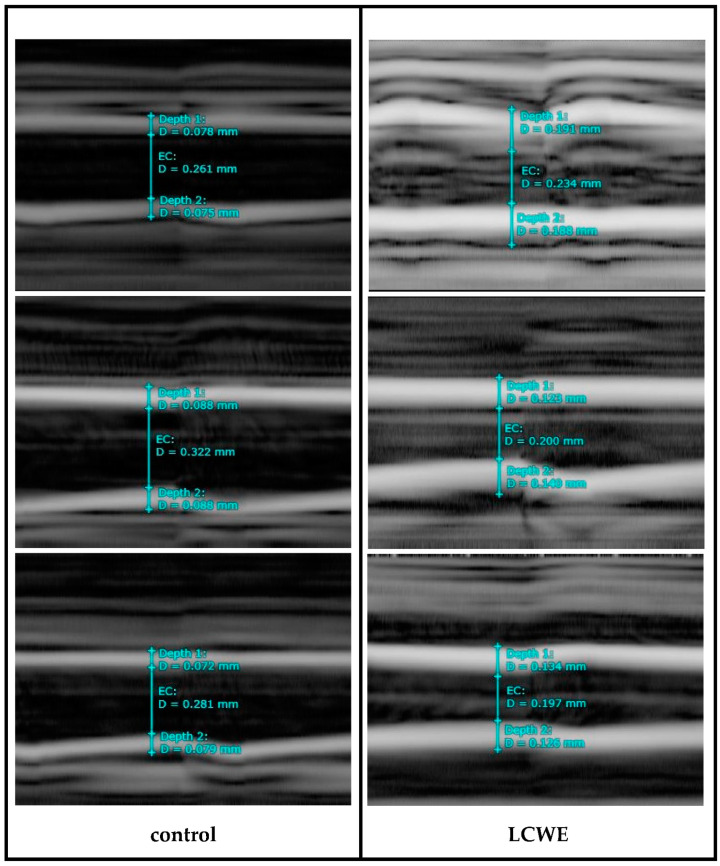
Carotid artery diameter and wall thickness in the control group (no LCWE treatment) and LCWE group (intraperitoneal injection 2 mg/kg for 7 days) by sonography on day 28 after study. Wall thickness significantly increased in the LCWE group with the narrowing of the internal diameter (EC: carotid artery, D: diameter, Depth 1 and 2: wall thickness).

**Figure 4 diagnostics-14-00145-f004:**
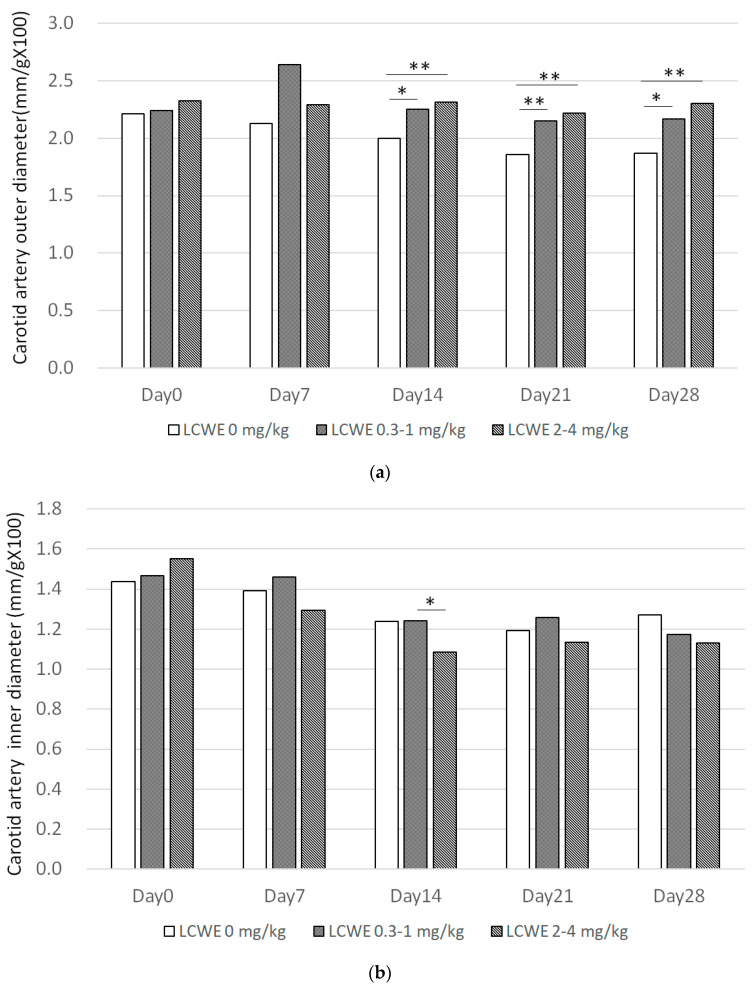
(**a**,**b**) Carotid artery diameter with different LCWE dosages and times. (**c**) Carotid artery wall thickness significantly increased with dose and was time dependent. Calculation formula = upper wall of carotid artery wall + lower wall divided by 2 divided by body weight multiplied by 100. (* *p* < 0.05, ** *p* < 0.005, *** *p* < 0.0005).

**Figure 5 diagnostics-14-00145-f005:**
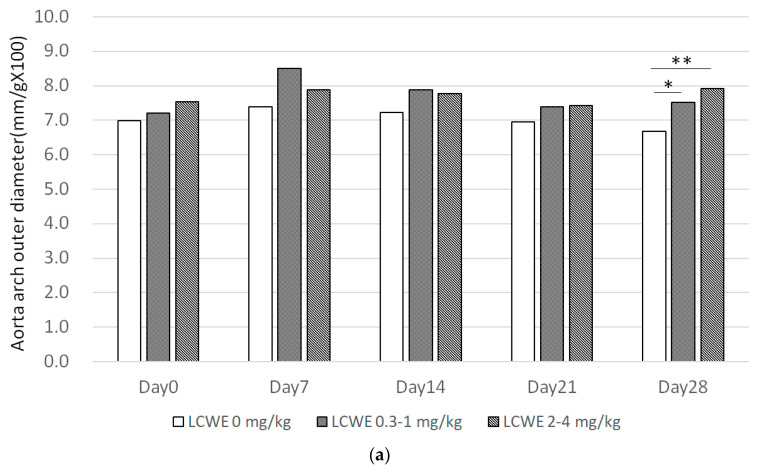
(**a**) Aorta outer diameter with different LCWE dosages and time showed a significant increase on day 28. (**b**) Aorta inner diameter with different LCWE dosages and time showed a significant increase on day 28. (**c**) Aortic arch wall thickness significant increased in LCWE group. Calculation formula = upper + lower aortic wall divided by 2, divided by body weight multiplied by 100. (* *p* < 0.05, ** *p* < 0.005).

**Figure 6 diagnostics-14-00145-f006:**
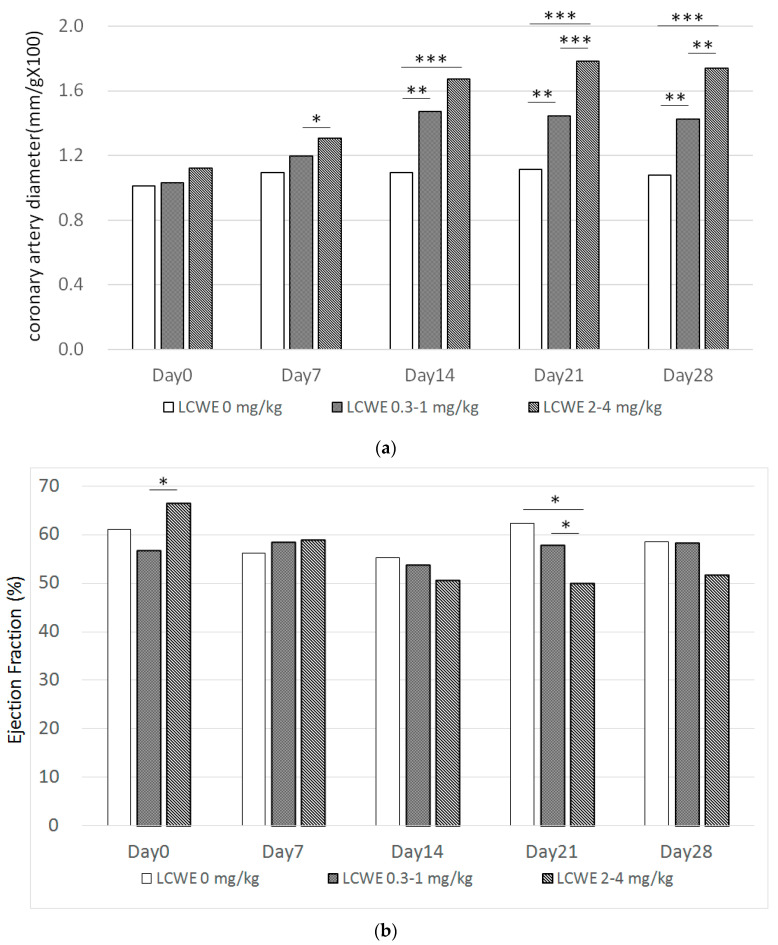
(**a**) Coronary artery diameter with different LCWE dosage and time showed a significant increase after day 7. (**b**) Ejection fraction showed significant decrease in LCWE group. (* *p* < 0.05, ** *p* < 0.005, *** *p* < 0.0005).

**Figure 7 diagnostics-14-00145-f007:**
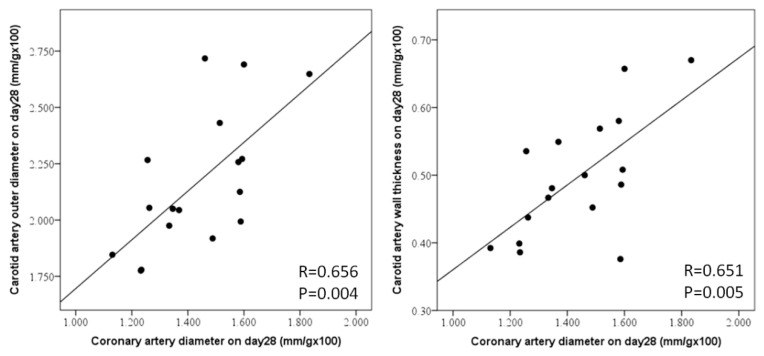
Carotid artery outer diameter and wall thickness were positively associated with coronary artery diameter on day 28. Calculation formula = upper wall of carotid artery wall + lower wall divided by 2 divided by body weight multiplied by 100.

**Figure 8 diagnostics-14-00145-f008:**
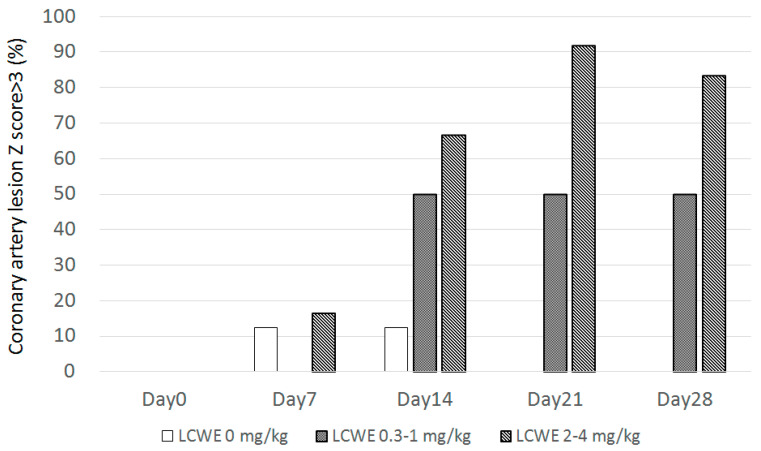
The percentage of *Z* > 3.0 at different dosages and days.

## Data Availability

The data that support the findings of this study are available from the corresponding authors.

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
