# Peer review of "Positive Echocardiographic Association between Carotid Artery and Coronary Artery Diameter and Z-Score in a Mouse Model of Kawasaki Disease"

_diagnostics, 2024, doi:10.3390/diagnostics14020145_

Round 1

Reviewer 1 Report

Comments and Suggestions for Authors

Dear Authors, Dear Editors,

I read Kuo's work with interest, which addresses a relevant question in a work that is not least completely enlightened. The introduction leads to the question. In my view, the methods and results need to be supplemented. On the one hand, clarity about the exact number and the course or the measurements would have to be presented, e.g. by means of a flow chart, since it is not entirely clear which measurement is measured for which animal. It would also be desirable to have a more detailed description of the actual measurement method and echocardiography. In the results section, detailed characteristics and information on the results would be desirable.

Author Response

Reviewer 1:

I read Kuo's work with interest, which addresses a relevant question in a work that is not least completely enlightened.

-->Thank you.

The introduction leads to the question. In my view, the methods and results need to be supplemented.

-->We have added in the results and methods.

On the one hand, clarity about the exact number and the course or the measurements would have to be presented, e.g. by means of a flow chart, since it is not entirely clear which measurement is measured for which animal.

--> We have added a flow chart.

It would also be desirable to have a more detailed description of the actual measurement method and echocardiography.

--> we have added the detailed description of the actual measurement method and echocardiography in the method section.

In the results section, detailed characteristics and information on the results would be desirable.

--> we have added the description in the result section.

Reviewer 2 Report

Comments and Suggestions for Authors

 The aim of this study was to survey the vasculitis status of the coronary artery and the carotid artery in a Kawasaki Disease mouse model. 

Method: LCWE was intraperitoneally injected into 5-week-old male C57BL/6 mice to induce CAL. We studied the longitudinal status of the carotid and coronary arteries, as well as analyzed the Z-score of coronary artery diameter. Results: Carotid artery wall thickness (day 7) and diameter (day 14) significantly increased in the LCWE group with a dose-dependent effect (p<0.05). Aortic diameter and wall thickness demonstrated significant increases on day 28 and day 7, respectively (p<0.05). Carotid artery outer diameter and wall thickness were positively associated with coronary artery diameter on day 28 (p<0.01). The percentage of Z > 3.0 indicated more than 80% in the high-dose LCWE group and 0% in the control group. 

Conclusions: This report is the first to use coronary artery Z-score in a mouse model of KD by echocardiography and find a positive association between carotid artery and coronary artery diameter.

this analysis was conducted limited to Kawasaki disease and was reproduced in a mouse model. Therefore, this evaluation should be performed in human subjects to see if an association is evident.

The article appears well written and is an enjoyable read.

I recommend a paragraph regarding the limitations of the study.

Comments on the Quality of English Language

Minor revision

Author Response

this analysis was conducted limited to Kawasaki disease and was reproduced in a mouse model. Therefore, this evaluation should be performed in human subjects to see if an association is evident.

--> we have add the human subjects data. (Wu et al. reported that mean carotid IMT in KD patients (0.550 ± 0.081 mm; range, 0.44–0.69 mm), was significantly higher than that in the febrile control children (0.483 mm± 0.046 mm; range, 0.43–0.56 mm; P = 0.01)) and mentioned in the limitation.

The article appears well written and is an enjoyable read.

I recommend a paragraph regarding the limitations of the study.

--> We have added a paragraph regarding the limitations of the study.